# Are the Post-COVID-19 Posttraumatic Stress Disorder (PTSD) Symptoms Justified by the Effects of COVID-19 on Brain Structure? A Systematic Review

**DOI:** 10.3390/jpm13071140

**Published:** 2023-07-15

**Authors:** Georgios D. Kotzalidis, Ottavia Marianna Ferrara, Stella Margoni, Valentina Ieritano, Antonio Restaino, Evelina Bernardi, Alessia Fischetti, Antonello Catinari, Laura Monti, Daniela Pia Rosaria Chieffo, Alessio Simonetti, Gabriele Sani

**Affiliations:** 1NESMOS (Neurosciences, Mental Health, and Sensory Organs) Department, Faculty of Medicine and Psychology, Sant’Andrea Hospital, Sapienza–Università di Roma, 00189 Rome, Italy; giorgio.kotzalidis@gmail.com; 2Department of Neuroscience, Section of Psychiatry, Fondazione Policlinico Universitario Agostino Gemelli IRCSS, 00168 Rome, Italy; ottaviaferrara@icloud.com (O.M.F.); stella.margoni98@gmail.com (S.M.); valentinaieritano.nutrizione@gmail.com (V.I.); restainoantonio11@gmail.com (A.R.); evelinabernardi@gmail.com (E.B.); alessiafischetti27@gmail.com (A.F.); catinariantonello@gmail.com (A.C.); gabriele.sani@unicatt.it (G.S.); 3Centro Lucio Bini, 00193 Rome, Italy; 4UOS Clinical Psychology, Clinical Government, Fondazione Policlinico Universitario Agostino Gemelli IRCCS, 00168 Rome, Italy; laura.monti@policlinicogemelli.it (L.M.); danielapiarosaria.chieffo@policlinicogemelli.it (D.P.R.C.); 5Women, Children and Public Health Department, Catholic University of the Sacred Heart, 00168 Rome, Italy; 6Menninger Department of Psychiatry and Behavioral Sciences, Baylor College of Medicine, Houston, TX 77030, USA; 7Institute of Psychiatry, Department of Neuroscience, Catholic University of the Sacred Heart, 00168 Rome, Italy

**Keywords:** neuroimaging, COVID-19, post-traumatic stress disorder, pandemic, hippocampus, anterior cingulated cortex

## Abstract

COVID-19 affects brain function, as deduced by the “brain fog” that is often encountered in COVID-19 patients and some cognitive impairment that is observed in many a patient in the post-COVID-19 period. Approximately one-third of patients, even when they have recovered from the acute somatic disease, continue to show posttraumatic stress disorder (PTSD) symptoms. We hypothesized that the persistent changes induced by COVID-19 on brain structure would overlap with those associated with PTSD. We performed a thorough PubMed search on 25 April 2023 using the following strategy: ((posttraumatic OR PTSD) AND COVID-19 AND (neuroimaging OR voxel OR VBM OR freesurfer OR structural OR ROI OR whole-brain OR hippocamp* OR amygd* OR “deep gray matter” OR “cortical thickness” OR caudate OR striatum OR accumbens OR putamen OR “regions of interest” OR subcortical)) OR (COVID-19 AND brain AND (voxel[ti] OR VBM[ti] OR magnetic[ti] OR resonance[ti] OR imaging[ti] OR neuroimaging[ti] OR neuroimage[ti] OR positron[ti] OR photon*[ti] OR PET[ti] OR SPET[ti] OR SPECT[ti] OR spectroscop*[ti] OR MRS[ti])), which produced 486 records and two additional records from other sources, of which 36 were found to be eligible. Alterations were identified and described and plotted against the ordinary PTSD imaging findings. Common elements were hypometabolism in the insula and caudate nucleus, reduced hippocampal volumes, and subarachnoid hemorrhages, while white matter hyperintensities were widespread in both PTSD and post-COVID-19 brain infection. The comparison partly supported our initial hypothesis. These data may contribute to further investigation of the effects of long COVID on brain structure and function.

## 1. Introduction

The Coronavirus Disease-2019 (COVID-19) pandemic, first declared on 11 March 2020 [1], has been associated with 687,021,745 infections worldwide up to now and 6,863,517 deaths (approximately 1% mortality), with deaths peaking and plateauing between April 2020 and April 2022 and new cases peaking between January and April 2022 [2]. A vast majority of patients infected with the Coronavirus recovered, but some of them—in the range of 6.2% [3] to 45% [4]—go on to develop long COVID. Of them, approximately 30% develop frank posttraumatic stress disorder (PTSD) or PTSD symptoms [5,6], but figures of approximately 50% have also been reported [7]. We suspected that the coronavirus somehow induces brain morphological and functional abnormalities that could be similar to those that patients with PTSD exhibit upon neuroimaging. These could either precede (creating vulnerability towards the development of PTSD) or follow the establishment of the disorder. To date, some specific brain alterations have been shown to accompany PTSD. Brain structure and function alterations have been hypothesized to occur after exposure to traumatic exposure and condition subsequent brain maturational processes, thus pointing to a neurodevelopmental origin of trauma-related pathology [8]. PTSD youths were shown to significantly increase hippocampal activation in response to threatening images compared to typically developing youth; also, patients with pediatric PTSD with a remitting condition show increasing functional connectivity (FC) between the hippocampus and visual cortex while viewing threat stimuli. The increased hippocampal activation in response to threat and the decreased FC in the hippocampal–visual cortex 4 network could be one of the reasons why PTSD persists in a pediatric population [9]. Persistence in youth could be also attributable to atypical insular neurodevelopment since the insula is expected to increase across development and consequent brain maturation; the failure to do so, as shown with longitudinally employed magnetic resonance imaging (MRI), renders the affected people resistant to treatment [10].

Multiple factors have concurred with the determination of the traumatic impact of contracting COVID-19 [11], and social factors like the presentation of the pandemic by social and mass media cannot be ignored [12]. Although much work has focused on demonstrating the traumatic meaning of contracting COVID-19 and long COVID [13,14], to date, no study has focused on the neuroimaging cross-section of trauma and COVID-19. Establishing the existence of a common cross-section between the neuroimaging underpinnings of trauma and COVID-19 would pave the way to treatments sharing common elements in dealing with both groups of patients.

### 1.1. Studies of the Effects of PTSD on Brain Structure and Function 

Other nuclei that have been involved with the development of PTSD are the amygdala, hippocampus, rostral anterior cingulate cortex (ACC), and ventromedial prefrontal cortex (PFC). Results generally indicate that smaller hippocampal volumes are associated with PTSD [15], but it is not known whether this is a cause or effect (it may well be both); the evidence for rostral ACC, ventromedial PFC, and amygdala is less clear or strong [16]. Youths with PTSD showed reduced gray matter (GM) volumes in the right ventromedial PFC and bilateral ventrolateral PFC, whereas they showed increases in the dorsolateral PFC; this is different from typically developing youths, who show maturation-related decreases in dorsolateral PFC GM volumes with decreased PFC–amygdalar FC and PFC–hippocampal FC [17]. Such findings may relate to alterations that are not specific to PTSD but are shared by other comorbid psychiatric disorders [18].

We especially focused on the neuroimaging findings of patients with neurological symptoms of any origin. Neurological symptoms may not be the result of a viral invasion of the brain or viral replication in the brain [19], but they are compatible with immunological reactivity [20], possibly due to an “original antigenic sin” [21] leading to “immunological imprinting”, i.e., to the tendency of the immune system to respond to antigenic challenges the same way it responded previously (this holds true for any biological system and constitutes the core element of learning and adaptive responses); this could lead to increased antibodies against other viruses and lower responses to COVID-19 [22], something that has been observed in some individuals with COVID-19 [23,24]. It has been recently supported that the original antigenic sin could be responsible for shaping the humoral immune response to COVID-19 and may be related to the development of neurological symptoms in patients with long COVID [25]. Immune dysregulation in COVID-19 could lead to the commonly observed microbleeding and endotheliitis that involve various organs, including the brain [26]; the finding of immunologically-induced endothelial alterations is universal in the brains of patients who died during their COVID-19 infection [27].

### 1.2. Aim

Having hypothesized that the brain alterations that are associated with PTSD would overlap with COVID-19–induced brain alterations, we were prompted to seek literature reporting on the neuroimaging findings of patients who had recently suffered from COVID-19. We here systematically review this evidence.

## 2. Materials and Methods

To review systematically the neuroimaging findings of the brain alterations in patients who either currently or previously had COVID-19, we carried out a PubMed search using the following search strategy: ((posttraumatic OR PTSD) AND COVID-19 AND (neuroimaging OR voxel OR VBM OR freesurfer OR structural OR ROI OR whole-brain OR hippocamp* OR amygd* OR “deep gray matter” OR “cortical thickness” OR caudate OR striatum OR accumbens OR putamen OR “regions of interest” OR subcortical)) OR (COVID-19 AND brain AND (voxel[ti] OR VBM[ti] OR magnetic[ti] OR resonance[ti] OR imaging[ti] OR neuroimaging[ti] OR neuroimage[ti] OR positron[ti] OR photon*[ti] OR PET[ti] OR SPET[ti] OR SPECT[ti] OR spectroscop*[ti] OR MRS[ti])). The choice of the above search strategy was based on the need to be omnicomprehensive; all authors added their expertise to refine the search. Inclusion criteria were having performed structural neuroimaging and reporting data. MRI (either voxel-based morphometry (VBM) or regions-of-interest (ROI) approaches), positron emission tomography (PET), single-photon emission computerized tomography (SPECT), and computerized tomography (CT) were considered appropriate. Functional MRI (fMRI) or FC studies were considered when the aforementioned techniques were involved, but they were excluded if they included only examination. Excluded were also case reports or case series; opinion papers, such as editorials or letters to the editor; comments on other articles; duplicates; article corrections referring to an article that was already present in the search; articles unrelated to what we were searching for; unfocused or inadequate designs (inadequate for our purposes); animal or in vitro studies; papers not reporting on COVID-19 patients; articles of side effects of COVID-19 vaccination (labeled as unfocused); protocols, which usually do not report data but pave the way for future studies (however, if reporting preliminary data, they were taken into consideration); exclusively *post-mortem* studies (but if they reported on brain pathological findings of patients dying during their COVID-19 infection, they were considered and included); studies containing no neuroimaging data; and reviews, but the latter were downloaded and hand-searched for possible additional eligible references that could have eluded our search.

After completing the search, we labeled all resulting records according to whether they were to be included or excluded (Appendix A). The principal reason for exclusion was provided for each article (Appendix A). In our review, we followed the 2020 PRISMA (Preferred Reporting Items for Systematic Reviews and Meta-Analyses) Statement indications [28]. In Figure 1, we display the PRISMA flow diagram. We provide the 2020 PRISMA Checklist in the Appendix A. Thus, the labels for exclusion were the following: Unrelated, Review, Unfocused, Case reports/series, No COVID-19, No neuroimaging, Unsuitable data presentation, Opinion, Protocol, Animal, Functional connectivity only, fMRI only, Duplicate, In vitro, and *Post-mortem*.

Three authors independently conducted the agreed search and compared their results. Eligibility was based on being an original study on patients who had contracted COVID-19 or who had a current COVID-19 infection, including humans, that provided data on patients’ neuroimaging. To establish eligibility for each study, all authors ran Delphi rounds until a full consensus was reached. All studies were downloaded with the exception of manifestly unrelated studies. Thereafter, all titles were introduced into our database where their characteristics were defined.

To assess the quality of eligible studies, we used the appraisal tool for cross-sectional studies (AXIS) [29] on each cross-sectional included study. The results of the assessment are shown in Appendix A.

There were no ethical concerns to address or ethical committee approval to obtain. The authors adhered to the principles of the WMA Helsinki Declaration of Human Rights and its subsequent amendments in their handling of information related to the patients involved in the eligible studies.

## 3. Results

On 29 April 2023, our above PubMed search yielded 486 articles; 2 more were obtained from the references of the obtained literature. Eligible were 36 studies (Figure 1; Table 1) [30,31,32,33,34,35,36,37,38,39,40,41,42,43,44,45,46,47,48,49,50,51,52,53,54,55,56,57,58,59,60,61,62,63,64,65]. These studies found a multitude of brain alterations with and after COVID-19. Most frequent were hypometabolism in the frontal cortex, ACC, insula, and caudate nucleus performed with ^18^F-fluorodeoxyglucose-PET/computerized tomography (8 studies), hemorrhages (nontraumatic subdural hemorrhages, nonaneurysmal subarachnoid hemorrhages, microhemorrhages, large parenchymal hemorrhage) (7 studies), hypoxic changes (7 studies), and supratentorial, middle cerebellar peduncular subcortical, periventricular, and deep white matter lesions (7 studies). Details are shown in Table 1.

Of the 36 eligible studies, the majority (N = 23) were single-site, while 11 were multicenter (5 2-center, 2 3-center, one each 4-, 5- and 6-center, and 1 11-center); one used BioBank data across the entire United Kingdom (UK) and another was an international call on behalf of the American Society of Pediatric Neuroradiology (ASPN), which recruited from 10 countries. As is currently fashionable, no multicenter study reported on intersite differences. The majority of studies were conducted in the United States (USA) (N = 9, while a tenth study participated in a 5-center/4-country study), 7 were conducted in France, 2 were conducted in Italy (plus 1 in the aforementioned 5-center/4-country study), 2 were conducted in Sweden, 2 were conducted in Brazil (plus 1 in the 5-center/4-country study), 2 were conducted in the UK, 2 were conducted in Turkey, 2 were conducted in China, 1 was conducted in Spain (plus 1 in the 5-center/4-country study), 1 was conducted in the Netherlands, 1 was conducted in Switzerland, and the international call of the ASPN resulted in recruitment from France, the UK, the USA, Brazil, Argentina, India, Peru, and Saudi Arabia. Location did not affect study quality (AXIS assessment, Appendix A). Details about locations and correspondence with eligible studies are shown in Appendix A.

Neuroimaging in these studies was generally arranged according to patient needs and correctly started first with a computerized tomography (CT) scan, identifying patients with putative abnormalities, and progressing thereafter to more sophisticated and in-depth investigation methods like MRI and PET. MRI used more commonly 3Tesla instrumentation (6 studies) and less frequently 1.5Tesla apparatuses, indicating a transition in instrumentation use towards upgraded tools. We did not exclude articles including 1.5Tesla apparatuses as the studies using them still provide valid results. Seven studies used PET with 2-deoxy-2-[fluorine-18]fluoro-D-glucose, four studies combined it with MRI, and three used it alone. The studies included here [30,31,32,33,34,35,36,37,38,39,40,41,42,43,44,45,46,47,48,49,50,51,52,53,54,55,56,57,58,59,60,61,62,63,64,65] identified brain alterations in a subset of patients—those with identifiable alterations—and classified them roughly, without focusing on GM- or WM-specific alterations and brain nuclei/area volumes. In contrast, studies on brain consequences or correlates of PTSD [8,9,10,15] were more detailed as concerns structural and functional alterations in patients with PTSD. The findings of the studies included in this review only partially overlapped with those already reported in the literature for PTSD. Therefore, we started from the reported alterations in the brains of patients with COVID-19, attempting to identify such alterations in the PTSD literature. The evidence will be presented in the discussion that follows.

## 4. Discussion

In this review, we included studies reporting brain alterations in patients with recent or current COVID-19 infections; these included patients with long COVID, whom we expected to manifest PTSD symptoms. However, just one study reported PTSD symptoms [50], probably due to the lack of application of instruments that could reliably identify PTSD symptoms in most studies. The most frequently reported alterations were hypometabolism in the frontal cortex, ACC, insula, and caudate nucleus; hemorrhages, which all too frequently were microbleeds; hypoxia; and supratentorial, middle cerebellar, peduncular subcortical, periventricular, and deep white matter lesions. Surprisingly, there was not much reference to the hippocampus (only 4 of the 36 eligible studies), but studies reporting on the hippocampus identified hippocampal abnormalities. Results were not meta-analyzable due to extreme heterogeneity due to the variety of objectives and aims of each research team. Focusing on only methodologically consistent studies would have resulted in including a very reduced number of articles that would have been unreviewable.

By performing specific PubMed searches, we identified one study identifying frontal cortex hypometabolism in torture victims with PTSD [66] and two which found hypometabolism in the insula [66,67], the latter of which included one patient with PTSD following domestic violence and one finding moderate hypometabolism in the caudate nucleus [66]. The latter study also identified hippocampal volume reductions.

Subarachnoid hemorrhages are related to PTSD. People with a subarachnoid hemorrhage are more likely to develop PTSD in a Chinese population [68] and the subarachnoid hemorrhage population shows more PTSD than other populations, but the time course is different among individual patients [69]. Post-ictus patients have a >33% chance to develop PTSD [70,71]. However, there were no studies investigating radiologically present hemorrhages, macro-, or microbleeds in patients with PTSD. At any rate, the above studies do show a link between brain hemorrhages and PTSD, but they do not indicate the direction of this link. While it is probable that one-third of patients suffering these hemorrhages will go on to develop PTSD, it is not known whether patients with PTSD will develop intracranial hemorrhages.

Hypoxia is a factor in PTSD allowing memory loss of a traumatic event in PTSD patients [72]. Post-avalanche survivors develop PTSD in approximately 11% of cases and it is presumable, but not demonstrated, that suffering hypoxia is a factor in this development [73]. Most studies investigating hypoxia in PTSD were animal studies; the directionality of hypoxia in PTSD has not been demonstrated and no study has investigated radiological signs of hypoxia in PTSD patients.

Many studies investigated white matter integrity and identified diffuse hyperintensities in patients with PTSD. These studies employed more sophisticated techniques, like diffusion tensor imaging, while, among our eligible studies, few of them used it [50,60]. One study observed increased fractional anisotropy in multiple white matter tracts in patients with PTSD compared with controls subjected to trauma that did not develop PTSD [74]. Another study found reduced fractional anisotropy in patients with PTSD undergoing trauma-focused cognitive behavior therapy to correlate with dysphoric symptom reduction [75]. Still another study identified white matter abnormalities in patients with PTSD, i.e., reduced fractional anisotropy and increased radial diffusivity in white matter tracts like the corpus callosum, the external and internal capsules, cingulum, and inferior and superior longitudinal fasciculi [76], all findings shared also by the COVID-19 patients of our review. A role of trauma in white matter disruption may be suspected, inasmuch as both patients with PTSD and non-PTSD trauma-exposed individuals showed increased fractional anisotropy in white matter tracts like the anterior limb of the internal capsule, the forceps of the corpus callosum, and the corona radiata compared to a healthy control group [77]. Decreased baseline fractional anisotropy was confirmed by a Chinese study in the right cingulate gyrus, uncinate fasciculus, superior longitudinal fasciculus, corticospinal tract, inferior fronto-occipital fasciculus, inferior longitudinal fasciculus, and forceps major for parents having lost their only child, but it did not persist at follow-up, with PTSD progressively resolving [78]. Significant negative correlations between PTSD symptom severity and fractional anisotropy values were found in the left corticospinal tract and left inferior cerebellar peduncle [79], matching some of the results of our review. The most recent study on this subject reported reduced fractional anisotropy and increased radial diffusivity in patients with PTSD or mild traumatic brain injury who displayed psychological symptoms [80]. Making sense of all these findings, white matter alterations in PTSD match white matter alterations that accompany COVID-19.

### Limitations

This review has not been registered to PROSPERO. Furthermore, due to design heterogeneity, we could not perform meta-analyses or assess the risk of bias of each included study. However, we used the AXIS instrument to assess the quality of the cross-sectional studies we included [29] (although there were studies with longitudinal aspects, none was fully prospective longitudinal) and found the mean quality to be medium-high. Due to the restricted time of the COVID-19 pandemic and the recent time period involved in these studies, this was to be expected (i.e., recent studies are expected to be more methodologically sound). It should also be stressed that some of the alterations that we found to be shared among PTSD patients and survivors of COVID-19 infection (PFC, ACC, insula, and hippocampus) may also be found in patients with emotional disturbances, like anxiety and depressive disorders [81,82,83,84,85], and even in severe psychotic disorders, like schizophrenia [86]. With these disturbances and disorders all having a shared background with childhood trauma and adversity [87,88,89], it is almost impossible to disentangle what is specific to PTSD and COVID-19 and what could be nonspecific.

The lack of prospective longitudinal studies does not allow strong conclusions to be drawn about predictors and risk factors, but, in COVID-19 times, cross-sectional studies on the particular issue we examined were all that we could obtain. Despite these limitations, the included studies allowed us to draw some conclusions that could constitute the basis of future studies.

Summarizing, of the brain alterations we identified in COVID-19 patients, some match the alterations encountered in PTSD and for some others, which are the most frequently observed changes in this review, the evidence is weaker.

## 5. Conclusions

Taking together all the evidence of brain alterations in COVID-19 and in PTSD, they overlap only partially, thus partially backing our hypothesis. One-third of patients with COVID-19 who developed PTSD could likely find a neurobiological basis of COVID-19-induced brain alterations. Since neuroinflammatory and immune reactivity mechanisms were advocated for the neurological symptoms of COVID-19, studies focusing on neuroinflammation and immunity are warranted. Such studies, despite appearing in the late nineties, are still scarce. It is important to continue to study this issue since it could open-up new ways of treating both PTSD cases and people who have suffered COVID-19 by enforcing treatment programs shared by both populations.

## Figures and Tables

**Figure 1 jpm-13-01140-f001:**
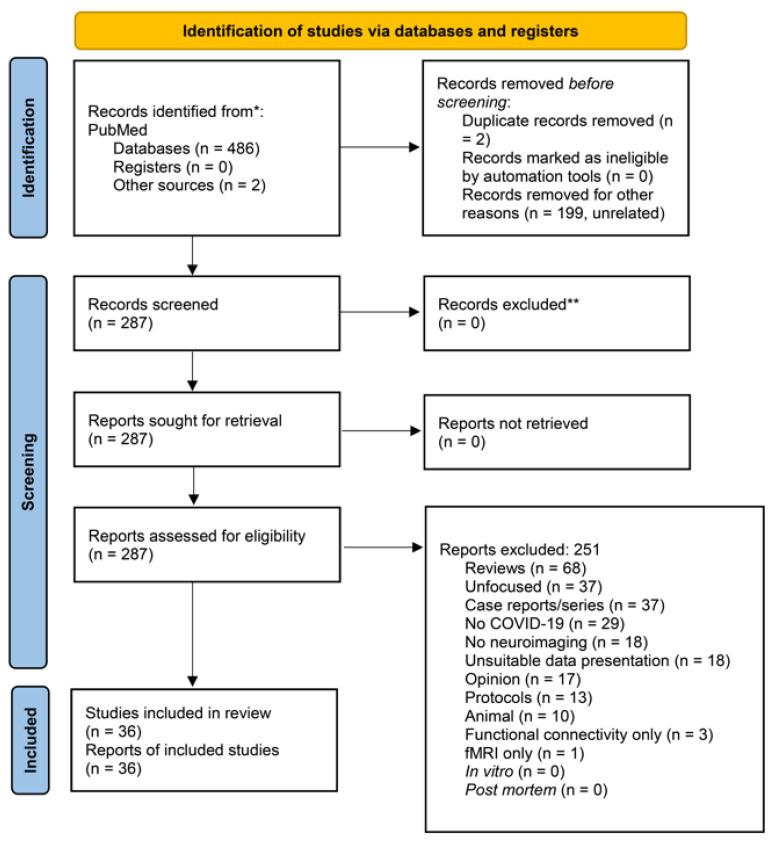
PRISMA-flow diagram of our search with exclusion criteria specified. PRISMA 2020 flow diagram for new systematic reviews which included searches of databases and registers only ((posttraumatic OR PTSD) AND COVID-19 AND (neuroimaging OR voxel OR VBM OR freesurfer OR structural OR ROI OR whole-brain OR hippocamp* OR amygd* OR “deep gray matter” OR “cortical thickness” OR caudate OR striatum OR accumbens OR putamen OR “regions of interest” OR subcortical)) OR (COVID-19 AND brain AND (voxel[ti] OR VBM[ti] OR magnetic[ti] OR resonance[ti] OR imaging[ti] OR neuroimaging[ti] OR neuroimage[ti] OR positron[ti] OR photon*[ti] OR PET[ti] OR SPET[ti] OR SPECT[ti] OR spectroscop*[ti] OR MRS[ti])) PubMed, 29 April 2023, 486 articles + 2 from other sources * Consider, if feasible to do so, reporting the number of records identified from each database or register searched (rather than the total number across all databases/registers). ** If automation tools were used, indicate how many records were excluded by a human and how many were excluded by automation tools. From: Page MJ, McKenzie JE, Bossuyt PM, Boutron I, Hoffmann TC, Mulrow CD, et al. The PRISMA 2020 statement: an updated guideline for reporting systematic reviews. BMJ 2021;372:n71. doi:10.1136/bmj.n71. For more information, visit: http://www.prisma-statement.org/.

**Table 1 jpm-13-01140-t001:** Summary of studies (in chronological order) on the neuroimaging of patients with COVID-19.

Study	Population	Technique	Design	Results	Conclusions/Observations
Kremer et al. 2020 [30]	43 ♂ and 21 ♁ COVID-19+ (age range 20–92, median age 66)	MRI 1.5T/3T, 3D T1-weighted spin-echo w/wo contrast-enhanced imaging, DWI, gradient echo T2/susceptibility-weighted imaging, 2D/3D FLAIR postcontrast, 3D TOF MRA of the circle of Willis	Retrospective multicenter study. COVID-19^+^ pts. with neurological manifestations subjected to MRI	36 (56%) abnormal brain MRIs; most frequent neuroimaging findings: ischemic strokes (27%), LME (17%), and encephalitis (13%)	Pts. with COVID-19 may develop many a neurologic symptom. Immunological and neuroinflammatory mechanisms are supported by signs of inflammation in both CSF and neuroimaging
Klironomos et al. 2020 [31]	185 COVID-19+ pts. (138 ♂ and 47 ♁)	Brain CT w/wo contrast agent in standard doses, brain and spinal MRI 3T, SWI, T1-T2, FLAIR, DWI	174 pts. underwent 222 brain CT scans; 47 brain MRI scans performed in 43 (22 were contrast agent enhanced); follow-up MRI in 4 pts.; 7 pts. underwent spinal MRI (4 contrast-enhanced). Asymptomatic/mildly symptomatic pts. with COVID-19 conducted two brain MRI scans	Most common finding in pts. who underwent MRI, IA susceptibility abnormalities (74%), often with an ovoid shape suggestive of microvascular pathology and with a predilection for the CC (59%) and juxtacortical areas (36%). Ischemic and macrohemorrhagic manifestations observed. 44% pts. had leukoencephalopathy and 1 a cytotoxic CC lesion. Other findings: olfactory bulb signal abnormalities (19%), prominent optic nerve subarachnoid spaces (56%), and parenchymal enhancement (15%), LME (15%), cranial nerves (10%), and spinal nerves (50%). At follow-up MRI, leukoencephalopathy regressed and progressive LME emerged	Pts. with COVID-19 showed widespread vascular and inflammatory involvement of both central and peripheral nervous systems
O’Shea et al. 2021 [32]	308 hospitalized adult COVID-19+ pts. (179 ♂ and 129 ♁, median age 59.6); 142 underwent neuroimaging and 37 were + for coagulopathy	CT-A or contrast-enhanced CT, MRI or MRA, Doppler ultrasound	Adult pts. with COVID-19 had their demographic, hematologic, CS imaging, and clinical outcome (death and intubation) data collected. Imaging inspected for coagulopathy. Possible associations betwixt pt. demographics, blood markers, and outcomes sought using multivariable logistic regressions	142/308 (46%) pts. underwent 332 cross-sectional imaging, 37 of whom (26%) were coagulopathy^+^. Coagulopathy^+^ imaging consisted in pulmonary embolus (*n* = 21) (assessed with contrast-enhanced CT or CT-A), clot in upper- or lower-extremity veins (*n* = 13) (assessed with Doppler ultrasound), end-organ infarction in bowel (*n* = 4) and kidney (*n* = 4) (contrast-enhanced CT), and clot or parenchymal brain infarction (*n* = 2) (contrast-enhanced CT-A or MRI with MRA). Among coagulopathy^+^ pts., 8 (22%) had multisite involvement. No variable was significantly associated with coagulopathy^+^ imaging	Coagulopathy imaging manifestations commonly observed in hospitalized COVID-19^+^ pts. >⅕ of pts. with coagulopathy show multisite involvement. Clinical variables poorly predict which pts. will have + imaging, indicating the usefulness of complementing clinical assessment with imaging to detect COVID-19–associated coagulopathy
Lin et al. 2020 [33]	2054 adult, COVID-19+ pts. (age range 18–101 y, median age 64 years, 43% ♁ and 57% ♂); 278 underwent neuroimaging	MRI 3T, DWI, SWI, T1-weighted, T2 FLAIR, CT	Of 2054 adult COVID-19+, cross-sectional neuroimaging of the brain was performed for 278 (14%) pts., with 269 (13%) pts. undergoing CT, 51 (2.5%) pts. MRI, and 42 (2.0%) pts. both CT and MRI. For 17 of 51 pts. subjected to MRI, imaging was performed both before and after IV gadobutrol enhancement	Among those 278 pts. having neuroimaging data, 58 (21%) had + acute/subacute findings: 31 (11%) cerebral infarctions, 10 (3.6%) parenchymal hematomas, 6 (2.2%) cranial nerve abnormalities, 3 (1.1%) each PRES, probable CIAM, and nontraumatic subdural hemorrhages, while 2 (0.7%) had nonaneurysmal subarachnoid hemorrhages. ↑ yield of neuroimaging for pts. performing MRI (*n* = 51) with 26 (51%) showing acute/subacute findings	Variety of neuroimaging findings in COVID-19, such as ischemic strokes and intracranial hemorrhages, microhemorrhages with a predilection for the CC and olfactory nerve abnormalities
Sawlani et al. 2021 [34]	3403 adult, COVID-19 pts; 167/3403 pts. with neurological signs or symptoms requiring brain imaging	MRI (*n* = 36) 1.5T, T1-weighted sagittal, axial T2-weighted, FLAIR, SWI and DWI, CT (*n* = 172)	Of 3403 pts. with COVID-19, 167 (4.9%) had neurological signs/symptoms needing neuroimaging and were included in the study. Most common indications: delirium (44/167, 26%), focal neurological symptoms (37/167, 22%), and altered consciousness (34/167, 20%)	Neuroimaging abnormalities in 23% of pts. Abnormal MRI = 20/36; abnormal CT = 18/172. Main findings: microhemorrhages (*n* = 12), watershed WMHs (*n* = 4), SWI susceptibility changes in superficial veins (*n* = 3), acute infarct (*n* = 3), subacute infarct (*n* = 2), acute hemorrhagic necrotizing encephalopathy (*n* = 2), large parenchymal hemorrhage (*n* = 2), subarachnoid hemorrhage (*n* = 1), hypoxic-ischemic changes (*n* = 1), and ADEM-like changes (*n* = 1)	Varying imaging patterns on MRI: acute hemorrhagic necrotizing encephalopathy, WMHs, hypoxic-ischemic changes, ADEM-like changes, and stroke. Microhemorrhages were the most common finding (60% of pts, all showing CC splenial microhemorrhage)
Lindan et al. 2021 [35]	38 SARS-CoV-2 infection-related neurological symptoms + children. Participants included 13 from France, 8 from the UK, 5 from the US, 4 each from Brazil and Argentina, 2 from India, and 1 each from Peru and Saudi Arabia	MRI T1- and T2-weighted, DWI, FLAIR, CT	International call for cases of children with encephalopathy related to severe acute respiratory SARS-CoV-2 infection and abnormal neuroimaging findings. Clinical history and plasma/CSF data requested; neuroimaging data collected	Most common imaging patterns were postinfectious immune-mediated ADEM-like brain changes (16 pts.), myelitis (8 pts.), and neural enhancement (13 pts.). Cranial nerve enhancement could occur without corresponding neurological symptoms. Splenial lesions (7 pts.) and myositis (4 pts.) predominantly in children with multisystem inflammatory syndrome. Cerebrovascular complications rarer than in adults. Favorable outcome in most children	Children showed acute- and delayed-phase SARS-CoV-2-related CNS abnormalities. Recurring patterns of disease and atypical neuroimaging manifestations should be suspected as potentially due to SARS-CoV-2 infection
Orman et al. 2021 [36]	20 children SARS-CoV-2+ (male/female, 12:8)	Head CTs (6 with and 11 without contrast), brain MRIs (8 stroke protocol, 3 w/wo, 6 without contrast, 2 MRVs, and 7 MRAs)	20 COVID-19 + children underwent MRI (26) and/or CT (n = 17), CSF, and blood testing	10% of pts. (*n* = 2) had acute neuroimaging findings: subarachnoid hemorrhage combined with posterior reversible encephalopathy syndrome in 1 pt. and a right-sided hippocampal T2-hyperintense signal alteration in another, possibly secondary to seizure activity	COVID-19-related neurologic involvement seldom found in children. 90% of pts. showed no SARS-CoV-2 infection-related acute neuroimaging alterations
Rapalino et al. 2021 [37]	27 pts. (20 ♂ and 7 ♁), $\overset{-}{x}$ age 63 yrs; 7/20 showed leukoencephalopathy with ↓ diffusivity	MRI 1.5 and 3 T, diffusivity, axial SWI, axial FLAIR, axial T1, sagittal MPRAGE, axial T2 BLADE. Following contrast: administration: axial T1 and sagittal MPRAGE	27 consecutive pts. SARS-CoV-2 + had brain MRI following ICU admission; 7 developed unusual leukoencephalopathy with ↓ diffusivity on diffusion-weighted MRI. The remaining pts. did not show such a pattern. The study compared clinical, laboratory, and neuroimaging findings between the groups	The ↓ diffusivity group had a significantly ↑ BMI (36 versus 28 kg/m^2^, *p* < 0.01). Pts. with ↓ diffusivity → more frequent acute renal failure and ↓ estimated GFR values at the time of MRI. Pts. with ↓ diffusivity also showed ↓ lowest hemoglobin $\overset{-}{x}$ values and ↑ serum Na^+^ levels within 24 h before MRI. The distribution of confluent, mostly symmetric, supratentorial/middle cerebellar peduncular WM lesions was strikingly and significantly (*p* < 001) reproducible in the ↓-diffusivity group	In a consecutive cohort of adult COVID-19^+^ ICU pts., severe COVID-19 leukoencephalopathy with ↓ diffusivity was associated with an abnormal brain WM lesion distribution pattern, including diffuse, confluent, mostly symmetric supratentorial/middle cerebellar peduncular lesions
Kas et al. 2021 [38]	7 pts. with variable clinical presentations of COVID-19–related encephalopathy; 32 HCs	MRI, ^18^F-FDG-PET/CT	All pts. underwent CSF analysis, EEG, brain MRI, and ^18^F-FDG-PET/CT (acute phase 1 and 6 months after COVID-19 onset). PET images were analyzed with voxel-wise and ROI approaches and compared with those of 32 HCs	Structural MRI showed no cerebrovascular disease or COVID-specific abnormalities except for 1 pt. with typical WM enhancement. All pts. consistently showed hypometabolism in a frontal cortex-ACC-insula-CN neural network; 6 months post-COVID-19 onset, clinical improvement observed in most pts., but cognitive and emotional disorders persisted, accompanied by long-lasting prefrontal, insular, and subcortical ^18^F-FDG-PET/CT abnormalities	Fronto-cortical-ACC-insular-CN circuitry involvement could underlie the clinical features in pts. with COVID-19. The study suggests persistence of mild-to-severe impairment of the circuitry 6 months following COVID-19 infection
Conklin et al. 2021 [39]	16 pts., 5 ♁, 11 ♂, with severe COVID-19+	MRI 3 T, SWI, T1-T2, FLAIR, DWI. Pt.1: brain autopsy, microscopic analysis of brain with LH&E staining; IHC: neurofilament, CD3, CD163, CD68; RT-qPCR of SARS-CoV-2 in CSF	Pts. with severe COVID-19 underwent MRI. Brain autopsy, microscopic analysis of brain, and SARS-CoV-2 CSF PCR were performed for pt.1 who died	4/16 pts: multiple clustered lesions in CC. 4/16 pts: subcortical, periventricular, and deep WM. Pt.1: MRI hypointense foci, subcortical, and deep WM, i.e., CC, internal capsules, cerebellar WM, diffuse DWI, and WM hyperintensity. Autopsy: edematous brain with herniation of uncus and tonsillae, diffuse discoloration and punctuate hemorrhages in cortical GM junction and deep WM. Microscopic analysis: microhemorrhages and microscopic ischemic lesions. IHC: microglia and macrophage accumulation	Common cerebral microvascular lesions in severe COVID-19 pts. with neurologic deficits; hemorrhagic and ischemic etiologies both involved
Guedj et al. 2021 [40]	35 pts. with long COVID, $\overset{-}{x}$ age 55.01 ± 11.22 yrs, 20 ♁, 15 ♂, vs. 44 HC	^18^F-FDG brain PET scans at resting-state by PET/CT GE camera after IV administration of 150 MBq ×15-min acquisition 30 min post-injection	Retrospective collection of sociodemographic and clinical data. Whole-brain statistical analysis was performed at voxel-level with the SPM8 software to compare pts. with long COVID-19 to HCs	Pts. with long COVID-19 presented with significant hypometabolism in bilateral rectal/orbital gyrus, including the olfactory gyrus, right temporal lobe, amygdala and Hipp, and extending to the right thalamus, bilateral pons/medulla brainstem, and bilateral cerebellum	A profile of brain PET hypometabolism in long COVID patients was found, involving the olfactory gyrus and its connected limbic/paralimbic regions, extended to the brainstem and the cerebellum
Büttner et al. 2021 [41]	34 hospitalized COVID-19 pts., 26 ♂, 8 ♁, $\overset{-}{x}$ age 67.5 ± 17.6 yrs	Unhenanced CT, MRI, T1-T2, FLAIR, DWI, SWI, MRA	Retrospective analysis of brain CT and MRI scans of 34 hospitalized COVID-19 pts. Collection of clinical parameters such as neurological symptoms, comorbidities, and type of ventilation therapy	All pts. with pathological findings were intubated or oxygenated with ECMO at the time of CT/MRI; 26.5% pts. showed hemorrhagic manifestations: most commonly microbleeds, followed by focal sulcal convexity subarachnoid hemorrhage, superficial hemosiderosis of the convexity, *loco typico* hematoma, and lobar hematoma. Signs of hypoxic brain injury in 4 pts. (11.8%); acute or early subacute ischemic stroke in 2 (5.9%) pts. (1 cortical and 1 subcortical); generalized brain edema in 1 pt. (2.9%)	Pathological neuroimaging findings do occur in a substantial proportion of patients with severe COVID-19 disease needing intubation or ECMO
Thurnher et al. 2021 [42]	48 ICU pts. who underwent mechanical ventilation or ECMO	MRI, FLAIR, 3D-T1-weighted, coronal T2-weighted, DWI, SWI	Retrospective analysis of MRIs of 48 pts. who underwent mechanical ventilation/ECMO. Collection of clinical data (indication for mechanical ventilation, type of mechanical ventilation, laboratory and clinical findings, outcome, imaging, and clinical follow-up)	14 pts. with brain microsusceptibility changes were identified, with an identical pattern of multiple SWI hypointense foci, located at the interface between GM and WM, both in subcortical WM and surrounding nuclei in 13 of 14 (92.8%) pts. In 8/14 (57.1%) pts., SWI foci were infratentorial in cerebellar hemispheres. Affected were CC in 10 (71.4%), internal capsule in 5 (35.7%), and midbrain/pons in 6 (42.8%) pts. 3 had intracerebral hematoma, 1 pt. hypoxic-ischemic brain injury, 2 pts. imaging findings related to PRES, and 2 pts. infarcts	Distinct patterns of diffuse brain SWI susceptibilities in critically ill patients who underwent mechanical ventilation/ECMO. The etiology of these foci remains uncertain, but the association with mechanical ventilation, prolonged respiratory failure, and hypoxemia seem plausible explanations
Agarwal et al. 2021 [43]	21 critically ill COVID-19+ pts., 18 ♂, 3 ♁, $\overset{-}{x}$ age 63 yrs (IQR: 50-69)	16 pts: MRI 3 T, SWI. 5 pts: MRI 1.5 T, T2-weighted GRE	Retrospective review of BL and final MRIs to measure % change in bicaudate index and 3rd ventricle diameter and evaluate changes in the presence and severity of WM changes	Between BL and final MRI, the bicaudate index and/or 3rd ventricle diameter ↑ for 15 pts. (71%), 2 (13%) for bicaudate index and 2 (13%) for 3rd ventricle diameter. $\overset{-}{x}$ % change between BL and final imaging was 4.4% (IQR -3–20) for bicaudate index and 4.1% (0–25) for 3rd ventricle diameter. All 21 patients had WM changes on their BL MRI: 8 pts. (38%) were Fazekas 1, 7 (33%) Fazekas 2, and 6 (29%) Fazekas 3	On serial imaging of critically ill pts. with COVID-19, ventricle size frequently ↑. The varied evolution of WM changes suggests they were the result of both static and dynamic processes and that, while some WM changes are reversible, others are irreversible. There is probably a spectrum of pathophysiological processes responsible for these MRI brain changes
Mahammedi et al. 2021 [44]	135 hospitalized COVID-19+ pts. with acute neurological symptoms, 86 ♂, 49 ♁, $\overset{-}{x}$ age 68.2 ± 15.1 yrs	Chest and brain CT, MRI 1.5 T, Gd-DTPA for contrast studies. 6 scans with 3D-FLAIR and 3D T1-weighted postcontrast images	Retrospective review of imaging of hospitalized COVID-19+ pts	49 (36%) pts. had acute neuroimaging abnormalities and 86 (64%) pts. had nonacute neuroimaging findings. Pts. with acute neuroimaging abnormalities had significantly ↑ $\overset{-}{x}$ CT lung severity scores. 38 (28%) pts. had acute ischemic infarcts, 14 (10%) intracranial hemorrhage, and 22 (36%) abnormal WM. Of the ischemic infarcts, 21 (15%) were large, 10 (7%) were small/watershed, 5 (4%) were cardioembolic, and 2 (1%) were hypoxic-ischemic encephalopathy. Microhemorrhage was the most common intracranial hemorrhage followed by subarachnoid hemorrhage. The most frequent MRI findings of WM abnormality were nonconfluent punctate multifocal T2/FLAIR hyperintense lesions with associated microbleeds and confluent symmetric T2/FLAIR hyperintensity involving deep and subcortical WM	Pts. with COVID-19 with neurologic symptoms and acute abnormalities on neuroimaging had ↑ CT lung severity scores compared with patients with COVID-19 with neurologic symptoms but without acute neuroimaging findings
Lambrecq et al. 2021 [45]	57pts; age unspecified; sex unspecified of an original sample of 78 pts, $\overset{-}{x}$ age 61, 57 ♂, 21 ♁, COVID-19^+^	MRI 3 T w/wo Gd contrast	Retrospective cohort study, 78 COVID-19+ pts. enrolled; 57 of 78 underwent brain MRI	57 pts. with MRI; 41/57 had abnormalities; 13/41 had acute ischemic lesions; 5/41 showed WM-enhancing lesions; 4/41 with basal ganglia abnormalities; 3/41 had metabolic lesions; 19 pts. showed hypoperfusion	MRI findings showed perfusion and basal ganglia abnormalities, microhemorrhages, CC injury, and WM-enhancing lesions that may correlate to COVID-19 encephalopathy; limitations: small sample, no follow-up
Yan et al. 2021 [46]	5 pts; newborns; 2 ♂, 3 ♁ COVID-19^+^; 15 newborn HCs	MRI 1.5 T, T1- T2-weighted, DWI	Case–control; 5 COVID-19+ newborns (January–July 2020) and 15 matched HCs underwent MRI	4 of 5 pts. showed WM changes; 2 full-term pts. with hypoxic changes in the basal ganglia region; 1 pt and 1 full-term pt showed hypoplasia with delayed myelination; ↑ IFG, MFG, ROp, MTG, and precuneus on VBM in pt group. ↓ PHG, OFC, ACC, SFG, CN, and thalamic region on VBM in both groups. In pt. group, correlation between GMV and each part of the HNNE score; CN, PHG, and thalamus had stronger correlations with HNNE scores	Limitations: did not specify how many and which newborns were full-term; sample size
Uginet et al. 2022 [47]	39 pts; $\overset{-}{x}$ age 66.5 ± 9.2 yrs, 35 ♂, 4 ♁ COVID-19^+^	MRI 1.5 T w/wo Gd contrast	Retrospective observational study of 39 pts. who underwent brain MRI; 34 of 39 underwent the full protocol, including contrast-enhanced MRI	29 pts. enhancement of intracranial vertebral and basilar arteries on MRI images. 12/29 unilateral and 17/29 bilateral enhancement at the level of the vertebral arteries. On DWI, 8/39 pts. with acute ischemic stroke; on SWI, 23/39 pts. had microbleeds: 6/23 deep, 8/23 superficial, and 9/23 mixed microbleeds	COVID-19+ pts. presented Gd vessel enhancement mostly at basilar and intracranial vertebral arteries suggestive of endotheliitis. This COVID-19 encephalopathy may be secondary to vessel wall inflammation
Jegatheeswaran 2022 [48]	103 pts; $\overset{-}{x}$ age 73.3 ± 14.9 yrs, 56 ♂, 47 ♁ COVID-19+	CT, MRI 1.5 with or without Gd contrast	Observational study of 150 of 422 hospitalized pts. who underwent MRI; 103 were included. These pts. underwent 172 CT scans and 18 MRI scans	30 pts. were admitted to ICUs and 73 to other wards; 33 patients (whether with or without neuroimaging not specified) died during the study period. 4 of 17 with MRI scans showed SWI abnormalities; 1 pt with encephalopathy showed left parietal lobe sulcal asymmetry on nonenhanced MRI. Pts. hospitalized in ICUs were more likely to show acute neuroimaging abnormalities and macrohemorrhages, hence showing higher mortality	ICU pts. have more abnormal CT or MRI scans, with more anoxia and macrohemorrhages and abnormal SWI than pts. hospitalized in other wards; the paper also reported comorbidities, but failed to refer them to its specific subsamples
Tu et al. 2021 [49]	47 pts; $\overset{-}{x}$ age 51.8 ± 11.3 yrs, 14 ♂, 33 ♁ COVID-19+. 43 HCs, $\overset{-}{x}$ age 52.5 ± 11.0 yrs, 11 ♂, 32 ♁	MRI 3 T, T1 weighted fast spoiled GRE, T2-weighted GRE planar imaging, ALFF, fMRI	After discharge from hospital (February-March 2020), 50 pts. underwent MRI in August 2020 (3 pts. excluded due to MRI artifacts). 43 HCs performed MRI. Pts. and HCs were assessed with PCL-5	Pts. ↑ GM vol. in bilateral amygdala and hippocampus. Left amygdala and left hippocampus volumes negatively correlated with PCL-5. ALFF ↑ in bilateral amygdala and hippocampus in pts., compared to HCs	Both GM vol. and functional measures in bilateral amygdala and hippocampus were significantly greater in COVID-19 survivors. Limitations: short-medium follow-up; gender effect in MRI analysis not considered; incomplete exclusion of possible effects of COVID-19 infection and medications on brain abnormalities
Benedetti et al. 2021 [50]	42 pt., $\overset{-}{x}$ age 54.86 ± 7.89 yrs, 29; ♁, 13 ♂ COVID-19+	MRI 3 T, T1-T2 weighed FLAIR, DTI, fMRI, PCR of inflammatory markers, calculated SII	Pts. underwent MRI 90.59 ± 54.66 days after testing positive for COVID-19; BDI, ZSDS, IES-R 7d before MRI	↓ GM on VBM associated with psychopathological severity; BDI, ZSDS, and IES-R scores negatively correlated with GM in bilateral ACC (BA 24 and BA32); BDI and IES-R scores correlated negatively with GM in bilateral insula; IES-R was negatively associated with GM in the precuneus. WM microstructure alterations were in the same direction of SSI. IES-R negatively correlated with WM in both hemispheres, especially in superior and posterior corona radiate, SLF, ILF, external capsula, and anterior thalamic radiation; BDI was negatively associated with AD in left superior corona radiata, SLF, and posterior corona radiata	Both WM and GM and FC alterations may mediate the relationship between medical illness and psychopathological sequelae of COVID-19; the more wide associations of psychopathology and SII were with IES-R scores, underlining the importance of PTSD in COVID-19
Duarte et al. 2022 [51]	1359 pt., age range 53–79 yrs, 897 ♁, 462 ♂ COVID-19^+^	MRI 1.5 T, DWI, SWI, T1-T2-weighed FLAIR, TOF MRA, non-enhanced CT scans	COVID-19^+^ pts. assessed for required neuroimaging; 259 needed non-enhanced CT scan or MRI; 250 were excluded due to bad quality imaging (*n* = 2), chronic alterations unrelated to COVID-19 (*n* = 73), and no detectable alterations on CT (*n* = 175); 9 pts. underwent MRI 12 d after onset of respiratory symptoms	Ischemic stroke findings in right cuneus, right cerebellar hemisphere, pons, thalamus, left inferior parietal lobule, left superior frontal gyrus, cingulate gyrus, bilateral cerebellar hemisphere, brainstem, left inferior frontal gyrus, insula, right cingulate gyrus, bilateral posterior territory, left cerebellar hemisphere	Potential link between COVID-19 and cerebrovascular events. COVID-19 is related to stroke-like alterations in 0.66% of affected pts
Nelson et al. 2022 [52]	56 pt., age range, 23–79 yrs, 40 ♁, 16 ♂ COVID-19+, ICU survivors	MRI 3 T, chest X-ray and CT, T1-T2-weighed FLAIR, coronal STIR, REMyDI, ASL, 3D echo planar susceptibility-weighted, thoracic protocol MRI, EDSS, UPDRS, RAVLT, ROCF, VFT, A Category flow test, TMT, coding a subtest of WAIS-IV, digit span a subtest of WAIS-IV, MFS, RAND-36	ICU survivors March 2020–June 2021; 21 required mechanical ventilation, 3 needed NIV prior to ICU, 19 needed HFOC prior to ICU, and 4 needed tracheostomy. COVID-19^+^ pts. underwent neuropsychological assessment, self-reported questionnaires, and smell identification test. At 3- and 12-month follow-ups, pts. underwent MRI of brain and lungs, chest X-ray, and CT	2 pts. had incidental findings on brain MRI findings requiring activation of the Incidental Findings Management Plan. Several pts. expressed cognitive and/or mental concerns and fatigue, complaints closely related to brain fog	Potential link between ICU hospitalization and neurocognitive impairment after severe COVID-19
Andriuta et al. 2022 [53]	46 pts. with a post-acute COVID-19, $\overset{-}{x}$ age 50.9 ± 14 yrs, 11 ♁, 35 ♂, ↑ educational level	MMSE, BNT, ROCF, FCRST, doors and people test, DSCT, TMT, Stroop test, BDSI, MADRS, STAI, IADLs, MRI 1.5 T (sequences: 3D FLAIR, Gd 3D, T1-, T2-weighted gradient echo, diffusion)	The study included pts. with post-acute COVID-19+ cognitive complaints. They underwent a neuropsychological assessment and 36 had cerebral MRI. Time between COVID-19 and neuropsychological assessment was 254 d, time between COVID-19 and MRI was 202 d, time between MRI and neuropsychological assessment performed 54 d post-MRI	Cognitive deficit was slight and the cognitive domains most highly affected were action speed, executive function, and language (naming). Pts. with cognitive complaints presented WMHs, all right-sided and consisting of WMHs in the superior frontal region, postcentral region, right cingulum, cortico–spinal tract, ILS, internal capsule, and posterior segment of the arcuate fasciculus	The study demonstrates the presence of NCD in post-acute COVID-19 pts. with cognitive complaints, showing a predominance of slowing and executive dysfunction. The study confirmed the significant prevalence of NCD in post-acute COVID-19 syndrome and the importance of clinical follow-up of COVID-19 pts.
Widemon et al. 2022 [54]	The population of study was not described	Head NECT, head/neck CT-A, brain MRI w/wo, lumbar spine MRI without contrast	Inpatients underwent head NECT and head/neck CT-A. Outpatients underwent brain MRI w/wo and lumbar spine MRI without contrast. Retrospective weekly data were collected ×≈1 yr following WHO pandemic declaration (3 November 2020–3 September 2021) and compared to imaging volumes from previous year (3 November 2020–3 September 2021). Quarterly data were analyzed	↓ vol. head NECT persisting ×≈1 yr following WHO pandemic declaration; ↓ vol. head/neck CT-A, brain MRI w/wo, and lumbar spine MR without contrast during first quarter. Head/neck CT-A vol. returned to pre-pandemic levels by the 2nd quarter and ↑ above pre-pandemic levels during the 2nd and 3rd quarters. This finding may be attributable to a prothrombotic state in COVID-19 pts. Brain MRI w/wo and lumbar spine MRI without enhancement vols. returned to BL by the 2nd quarter	Both inpatients and outpatients showed ↓ head NECT vol. during the COVID-19 pandemic period, with inpatients taking longer to return to prepandemic vols. compared to outpatients
Lersy et al. 2022 [55]	31 pts; 74% ♂, 26% ♁ with prior severe COVID-19. $\overset{-}{x}$ age 61 ± 12.4 yrs, range 18-79 yrs	1.5-T MRI or a 3-T MR; 3D T1-weighted spin-echo MRI w/wo contrast enhancement; DWI, PWI, and SWI; 2D or 3D FLAIR before and after administration of gadolinium-based contrast agent. Resting-state ^18^FDG-PET/CT brain	Observational retrospective study. Between 1 March and 31 May 2020, 112 consecutive COVID-19 pts. with neurologic symptoms underwent brain MRI. 31 of 112 pts. underwent additional imaging study at 3 and/or 6 months and were then finally recruited. All 31 pts. were initially hospitalized in ICUs for severe disease. 23 pts. in this cohort underwent ^18^FDG-PET-CT after 3 months. 12 pts. underwent another ^18^FDG-PET-CT	Initial brain MRI findings: 32% normal; 45% focal (single focus or multiple foci) LME; 29% diffuse brain microhemorrhages, encompassing CC, subtentorial juxtacortical WM, internal capsule, brainstem, middle cerebellar peduncles, and cerebellum → diagnosis of CIAM; 13% acute ischemic strokes; 13% with arterial vessel wall thickening (vasculitis); 10% acute inflammatory demyelinating lesions (ADEM or AHL). Evolution at follow-up: LME—21% stability, 43% partial regression; 36% complete regression. Stability over time of CIAM. Normalization of vessel wall imaging. ADEM ensues in sequelae. Evolution of perfusion imaging: At first ASL brain perfusion imaging, 66.7% pts. had hypoperfusion; 17% had hyperperfusion. At follow-up, brain perfusion had normalized in 58%; 5% still had hyperperfusion; 5% with initial hyperperfusion had hypoperfusion at last imaging session; 37% pts. still had hypoperfusion at last imaging session. ^18^F-FDG PET/CT brain: The most affected regions were the temporal and insular regions (hypometabolism). 14 pts. had hypermetabolism in colliculi, especially at 3 months	This study showed stability or regression of lesions in most cases. On ^18^FDG PET/CT brain, pts. showed moderate hypometabolism, especially in temporal regions. Concerning LME, though they observed a declining trend, 9 out of 14 pts. (64.3%) still had this abnormality during follow-up. 3.2% ↓ of GM vol. during ≈5 months. Regarding CIAM, brain microbleed load was stable over time
de Paula et al. 2023 [56]	135 pts, 18–60 yrs; 25% ♂, 75% ♁ COVID-19+ in the last year; mild COVID-19, i.e., WHO clinical ordinal scale severity 1 and 2	3D T1, T2 and T2-FLAIR; T1; isotropic 3D T2-WI turbo spin-echo (SPACE); 3D FLAIR; DW-MRI; SWI; Resting-state ^18^F-FDG PET/CT brain	Prospective cohort study. Participants underwent 2 visits. 1st: neuropsychological assessment, neurological examination, and MRI. 2nd: blood tests and ^18^FDG-PET brain imaging	MRI: No structural changes in any of the 135 pts. No significant positive or negative correlations with scores on the ROCF at VBM-based analysis of GM images. Inverse relationship between ROCF copy performance and WM volumes encompassing the subgenual portion of CC and the cingulum on both hemispheres, WM portions of IFG and FOF bilaterally, and right fusiform gyrus and bilateral lingual gyri. ^18^FDG-PET: Negative correlation between ROCF copy performance and resting brain Gluc metabolism in frontal (right dorsal anterior cingulate gyrus, ROp, VLPFC, and left DLPFC) and occipital regions (bilateral IOG and left calcarine/lingual gyri); significant positive correlation involving left ITG and left IOG	↑ WM vol. related to visuoconstructional impairments affecting 25% of pts. in this study. Furthermore, resting brain Gluc metabolism in some frontal and occipital regions negatively correlated with visuoconstructional performances. Gluc metabolism in left ITG and left IOG positively correlated with visuoconstructional performances
Callen et al. 2023 [57]	15 pts, 4 ♂, 11 ♁; $\overset{-}{x}$ age 43 ± 12 yrs; mean time since infection 238 d	MRI 3T, anatomic T1-weighted 3D brain volume (BRAVO) sequence, ASL MR perfusion, VWI performed with a 3D high-resolution variable flip angle black-blood sequence, performed after IV administration of 0.2 mL/kg of Gd-based contrast medium	Perspective case- control study. 15 pts. with and 12 pts. without previous infection. Participants underwent MRI that included ASL perfusion imaging with acetazolamide stimulus to measure CBF and calculate CVR	Mean whole-cortex CBF after acetazolamide administration was greater in participants without previous infection. Whole-brain CVR was lower in participants with previous infections. CVR was lower in those with than those without post-COVID neurologic conditions, but this difference was not significant	Possible association between prior SARS-CoV-2 infection and impaired whole-brain and lobar CVR. No significant association between prior infection and presence of VWI abnormalities. Limitations: small sample size, MRI protocol did not include T2-weighted, FLAIR, or susceptibility-weighted sequences
Díez-Cirarda et al. 2022 [58]	86 pts. with PCS; $\overset{-}{x}$ age, 50.71 ± 11.20 yrs; 67.44% ♁	MRI 3T, resting-state fMRI, 3D T1-weighted images, sagittal 3D T2 FLAIR, DWI	Cross-sectional. 86 pts. with PCS and 36 HCs. Pts. underwent clinical and neuropsychological assessment and neuroimaging 11.08 ± 4.47 months since first symptoms of COVID-19	↓ FC between left and right PHG. ↓ FC from the left cerebellar III (vermis) to the left and right frontal superior orbital cortex. ↓ GM volume in the PHG, frontal gyrus, anterior cerebellar, occipital lobe, and bilateral superior temporal lobe. ↓ MD and AD in the CC, forceps minor, MLF, uncinate tract, and FOF; MD alterations mostly in the right hemisphere, while AD alterations bilateral in frontal (near the orbital area), temporal (next to the angular gyrus and PHG), parietal (next to precuneus), occipital and subcortical areas (proximal to the lentiform nucleus); GM atrophy significantly correlated with cognitive dysfunction	PCS patients presented hypoconnectivity between bilateral orbitofrontal areas and cerebellar area III (vermis) and between left and right PHG. They presented reduced ↓ AD and ↓ MD mostly lateralized to the right hemisphere in the following WM tracts: CC, forceps minor, SLF, inferior FOF, and uncinate tract. The combination of ↓ AD and ↓ MD may reflect axonal injury. The PHG region in PCS pts. showed FC alterations accompanied by GM vol. ↓ and presented adjacent WM abnormalities
Goehringer et al. 2023 [59]	28 PCC pts, $\overset{-}{x}$ age 46.1 ± 9.8 yrs; 25% ♂, 75% ♁	Resting-state ^18^F-FDG PET/CT brain	Retrospectively identified consecutive pts. who presented with PCC between September 2020 and May 2022 and had a brain ^18^F-FDG PET scan to investigate suspected brain involvement. All pts. underwent standardized clinical assessment (MoCA, HAD, mMRC, Chalder Fatigue scales). 28 age- and sex-matched HCs with no neuropsychiatric antecedents and normal neuropsychological tests from a local database	PCC pts. presented hypometabolic clusters predominantly located within the right frontal and temporal lobes, including the orbital and internal temporal areas. Brain hypometabolism mostly affected the right brain hemisphere. The brainstem and the cerebellum were not involved. No hypermetabolism was observed	^18^F-FDG PET showed a hypometabolism in the right fronto-temporal lobes. This study, differently from previous PET findings, found no involvement of the pons and cerebellar regions
Paolini et al. 2023 [60]	58 pts, COVID-19 survivors; $\overset{-}{x}$ age 52.34 ± 11.73 yrs; 41% ♂, 17% ♁	MRI 3.0 T with spin-echo-EPI, DTI T2-weighted, fMRI, MVPA	Cross-sectional perspective study. On the basis of their answers during the clinical interview, pts. were subdivided into cognitive noncomplainers (*n* = 29) and cognitive complainers (*n* = 29)	↑ MD bilaterally affecting the inferior FOF, uncinated fasciculus, and corona radiata as well as several CC sections. ↑ RD in several WM tracts located in the left hemisphere (corona radiata, ILS, inferior FOF, SLF, and uncinate fasciculus). ↑ AD in some inter-hemispheric associative tracts. ↑ FC in bilateral insular cortex, bilateral supramarginal and operular cortex, ACC, bilateral precentral gyrus, and inferior lateral occipital cortex; ↓ FC in bilateral superior occipital cortex, posterior cingulate gyrus, left MTG, and right cerebellum	Study showed ↑ MD in several bilateral WM tracts; ↑ of both AD and RD and a trend towards ↓ FA values in cognitive complainers. Abnormally ↑ resting FC in frontal pole with networks critically involved in cognitive-demanding tasks. All ↑ FC areas were part of the salience, sensorimotor, or dorsal attention networks (except for a small cluster in the inferior LOC belonging to the visual network); among clusters with ↓ FC, the three with the highest statistical significance were found to be part of the DMN
Kamasak et al. 2023 [61]	50 pts, COVID-19 survivors, 25 ♂, 25 ♁, vs. 50 HCs, 25 ♂, 25 ♁, age range 30–60	1.5 T MRI, T1-weighted 3D-MPRAGE, VBM	Participants underwent MRI and GM, WM, CSF, and total intracranial volume were calculated	Whole-brain GM vol. ↓ in post-COVID-19 vs. HCs. GM vol. in post-COVID-19 ↓ in gyri orbitales, gyrus rectus/BA 11-OFC, cingulate gyrus, pons, IFG, parietal lobe-BA7, supramarginal gyrus-BA 40, angular gyrus-BA 39, superior semilunar lobule-crus 1, Hipp, declivus vs. HCs. GM vol. of amygdala and WM volume of parietal lobe ↑ vs. HCs	COVID-19 negatively affects many CNS structures
Klinkhammer et al. 2023 [62]	101 pts. ICU COVID-19; $\overset{-}{x}$ age 61.0 yrs; 76 ♂, 25 ♁; 104 pts. non-ICU COVID-19; $\overset{-}{x}$ age 64.0 yrs; 68 ♂, 36 ♁	MRI 3T, T1-, T2-weighted FLAIR, DWI	Prospective cohort study. Of 1991 pts., 101 ICU and 104 non-ICU COVID-19+ were enrolled; 8-10 months post-hospital discharge, participants underwent neuropsychological testing and MRI	ICU pts. had ↑ microbleeds vs. non-ICU pts; N° of microbleeds significantly ↑ in the ICU group; microbleeds often in the CC	Despite ↑ microbleeds amongst ICU pts., cognitive dysfunction was equally present in both groups; ICU admission does not lead to worse cognitive functioning than other ward admission
Douaud et al. 2022 [63]	785 UK BioBank participants, age range 51–81 yrs, of which 401 COVID-19+ between scans (172 ♂ (42.9%), 229 ♁ (57.1%); $\overset{-}{x}$ age 58.9 ± 7.0 yrs (range 46.9-80.2 yrs)) and 384 HCs (164 ♂ 42.7%), 220 ♁ (57.3%) $\overset{-}{x}$ age 60.2 ± 7.4 yrs (range 47.1–79.8 yrs))	MRI scans (Tesla not specified) (T1-, T2- FLAIR), SWI, diffusion MRI, and resting-state and task fMRI	Used UK BioBank data ≈3 yrs apart; investigated regional GM, brain, and CSF vols.; local cortical surface area vol., and thickness; cortical GM-WM contrast; white matter hyperintensity volume; WMHs; FA; MD; resting-state amplitude; and dimensionally ↓ FC. Cognitive measures: TMT, SDT, VFT, reaction time, pair matching, maximum N° of digits recalled	Neuroimaging results: COVID-19 ↓ in GM thickness and tissue contrast in OFC and PHG vs. HCs; ↓ brain vol./estimated total intracranial vol.; ↑ CSF vol.; ↑ right lateral ventricle vol.; ↓ FC in temporal piriform network; ↓ superior FOF in the COVID-19 group vs. HCs. Cognitive results: TMT A and B took significantly longer for the COVID-19 group to complete than the HC group. In COVID-19, there was a significant longitudinal association with the vol. of the cerebellum’s mainly cognitive lobule-crus II	The COVID-19 group showed ↑ brain shrinkage and cognitive decline compared to HCs. Greater alterations observed in tissue damage markers in regions that are connected to the primary olfactory cortex; global brain size in the COVID-19 group parallels cognitive impairments and supports that COVID-19 negatively affects brain structure and function
Burulday et al. 2023 [64]	27 pts; $\overset{-}{x}$ age 35.25 ± 13.99 yrs; 16 ♂, 11 ♁; COVID-19; 27 HCs; $\overset{-}{x}$ age 35.62 ± 13.47 yrs; 16 ♂, 11 ♁	MRI 1.5 T, DWI	Retrospective study; 27 pts. COVID-19 and 27 HCs underwent MRI and blood sample	11 pts. smell loss, 16 pts. anosmia; thalamus bilaterally ADC ↓ in pts. Insular gyrus and corpus amygdala ADC: no differences between pts. and HCs	Restriction of diffusion in olfactory areas may be connected to damage at the neuronal level associated with COVID-19
Debs et al. 2023 [65]	45 pts; $\overset{-}{x}$ age 58 yrs (range 18–87 yrs); 24 (53.33%) ♂, 21 (46.67%) ♁; COVID-19; 52 pts. with melanoma or multiple myeloma; $\overset{-}{x}$ age 57 yrs (range 24–73 yrs); 28 (53.85%) ♂, 24 (46.15%) ♁	^18^FDG-PET/CT	Retrospective study; 45 pts. post-COVID-19, of whom 15 with previous PET and 52 malignancy pts. underwent ^18^FDG-PET/CT and SPM8 analysis	No differences between pre- and post-COVID-19 and controls in hypo- or hypermetabolism; extensive brain hypometabolism during the first 2 months post-onset of COVID-19 infection → progressive return to normal → 6–12 months ≈complete recovery of brain abnormalities with residual limited hypometabolic clusters in ACC, posterior IFG, right frontal operculum, and right temporal-insular region; hypometabolism disappeared at 12 months. COVID-19 vs. controls hypometabolism in bilateral parietal lobes-precuneus, frontal lobes (ACC-PFC), occipital lobes, right temporal lobe, and right cerebellum; in post-COVID-19, older age, neurologic symptoms, and severity positively correlated with degree of hypometabolism in bilateral parietal, posterior frontal, and temporal lobes and the degree of hypermetabolism in central cerebral and subcortical regions	Reversible brain PET hypo- and hypermetabolic changes in patients with COVID-19 infection. Alterations appear to be transient and positively correlated with older age, neurologic symptoms at the time of imaging, and worse disease severity. However, even asymptomatic patients showed metabolic changes on PET that strongly suggest COVID-19

Note. Abbreviations: 3D, three-dimensional; ^18^FDG-PET/CT, Positron emission tomography with 2-deoxy-2-[fluorine-18]fluoro-D-glucose integrated with computed tomography; ACC, anterior cingulated cortex; AD, axial diffusivity; ADC, apparent diffusion coefficient; ADEM, acute disseminated encephalomyelitis; AHL, hemorrhagic leukoencephalitis; ALFF, amplitude of low-frequency fluctuation; ASL, arterial spin labeling; BA, Brodmann’s area; BDI, Beck depression inventory; BDSI, behavioral dysexecutive syndrome inventory; BL, baseline; BLADE, motion correction with radial blades; BNT, Boston naming test; CC, corpus callosum; CIAM, critical-illness-associated cerebral microbleeds; CN, caudate nucleus; CNS, central nervous system; COVID-19, coronavirus disease 2019; CS, cross-sectional; CSF, cerebrospinal fluid; CT, computerized tomography; CT-A, computerized tomography–angiography; CVR, cerebrovascular reactivity; d, days; DLPFC, dorsolateral prefrontal cortex; DMN, default mode network; DSCT, digit symbol-coding test; DTI, diffusion tensor imaging; DWI, diffusion-weighted imaging; ECMO, extracorporeal membrane oxygenation; EDSS, expanded disability status scale; EPI, echo planar imaging; FA, fractional anisotropy; FC, functional connectivity; FCRST, free and cued selective reminding test; FLAIR, fluid-attenuated inversion recovery; fMRI, functional magnetic resonance imaging; FOF, fronto-occipital fasciculus; Gd, gadolinium; Gd-DTPA, gadolinium diethylene-triamine penta-acetic acid; GE, general electric; GFR, glomerular filtration rate; Gluc, glucose; GM, gray matter; GRE, gradient echo sequence; HAD, hospital anxiety and depression; head NECT, non-contrast enhanced computed tomography of the head; HC(s), healthy control(s); HFOC, high-flow oxygen cannula; Hipp, hippocampal, hippocampus; HNNE, Hammersmith Neonatal Neurologic Examination; IA, intra-axial, IADLs, instrumental activities of daily living scale; ICU(s), intensive care init(s); IES-R, impact of events scale—revised; IFG, inferior frontal gyrus; IHC, immunohistochemistry, immunohistochemical; ILS, inferior longitudinal fasciculus; IOG, inferior occipital gyrus; IQR, interquartile range; ITG, inferior temporal gyrus; IV, intravenous (not fourth; the latter should be 4th); LH&E, luxol and hematoxylin-and-eosin; LME, leptomeningeal enhancement; MADRS, Montgomery-Åsberg depression rating scale; MBq, mega-becquerel; MD, mean diffusivity; MFG, middle frontal gyrus; MFS, mental fatigue scale; MLF, middle longitudinal fasciculus; mMRC, modifed Medical Research Council; MMSE, mini mental state examination; MoCA, Montreal Cognitive Assessment; MPRAGE, magnetization-prepared rapid gradient-echo; MRA, magnetic resonance-angiography; MRI, magnetic resonance imaging; MTG, middle temporal gyrus; MVPA, multivariate pattern connectivity analyses; Na^+^, sodium ion; NCDs, neurocognitive disorders; NIV, non-invasive ventilation; N°, number; OFC, orbitofrontal cortex; PHG, parahippocampal gyrus; PCC, post-COVID-19 conditions; PCL-5, PTSD-Checklist for DSM-5; PCR, polymerase chain reaction; PCS, post-COVID syndrome; pt(s)., patient(s); PRES, posterior reversible encephalopathy syndrome; PTSD, posttraumatic stress disorder; PWI, perfusion-weighted imaging; QoL, quality of life; RAND-36, RAND Corporation 36-Item Health Survey to Measure QoL; RAVLT, Rey Auditory Verbal Learning Test; RD, radial diffusivity; REMyDI, rapid estimation of myelin for diagnostic imaging; ROCF, Rey–Osterrieth complex figure test; ROI(s), region(s)-of-interest; ROp, Rolandic operculum; SII, systemic immune inflammation index; RT-qPCR, real-time quantitative polymerase chain reaction; SDT, symbol digit test; SFG, superior frontal gyrus; SII, systemic immune inflammation index; SLF, superior longitudinal fasciculus; SPM8, statistical parametric mapping version 8; STAI, state-trait anxiety inventory; STIR, short tau inversion recovery; SWI, susceptibility-weighted imaging; T, Tesla; TMT, trail making test; TOF MRA, intracranial time-of-flight magnetic resonance angiography; UPDRS, unified Parkinson’s disease rating scale; VBM, voxel-based morphometry; VBM, voxel-based morphometry; VFT, verbal fluency test; VLPFC, ventrolateral prefrontal cortex; vol., volume; VWI, vessel wall imaging; w/wo, with and without contrast; WAIS-IV, Wechsler Adult Intelligence Scale Fourth Edition; WHO, World Health Organization; WMHs, white matter hyperintensities; WM, white matter; $\overset{-}{x}$, mean; yr(s), year(s); ZSDS, Zung self-rating depression scale; +, positive; ±, SD, standard deviation; ×, for, per; ≈, about equal, not different; ♁, females; ♂, males; ↓, decreased, lower; ↑, increased, higher, →, induced, followed.

## Data Availability

The data presented in this study are available in the article and Appendix A.

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
