# Peer review of "Are the Post-COVID-19 Posttraumatic Stress Disorder (PTSD) Symptoms Justified by the Effects of COVID-19 on Brain Structure? A Systematic Review"

_jpm, 2023, doi:10.3390/jpm13071140_

Round 1

Reviewer 1 Report

Thank you very much for giving me the opportunity to the review the manuscript entitled Are the post-COVID-19 posttraumatic stress disorder (PTSD) symptoms justified by the effects of COVID-19 on brain structure? A systematic review. In particular, authors examined wheter changes induced by COVID-19 on brain structure would overlap with PTSD. The topic is very interesting for readers, aims are clear and results are presented in a quantitative and structured way. All sections of manuscripts cover with the scope of the journal. However, some recommendations are suggested in order to enhace improvements in the current form of this manuscript.

Methods

1)      Did you register the review as a protocol -if not this needs to be added to your limitations.

2)      For the search strategy can you add how many people performed the search and if the processess were undertaken blind.

3)      Please consider the risk of bias, effect measures, reporting bias assessment and certainly assesment items with are in the PRISMA 2020 checklist.

Results

It would be betther if authors xould include the risk of bias assessment e.g AXIS tool for cross sectional studies and certainly assessment. It would improve the quality of results.

Despite this recommendation, from my point of view the core of the article is important and relevant and I suggest the paper has a huge potential for publication.

Author Response

Thank you very much for giving me the opportunity to the review the manuscript entitled Are the post-COVID-19 posttraumatic stress disorder (PTSD) symptoms justified by the effects of COVID-19 on brain structure? A systematic review. In particular, authors examined wheter changes induced by COVID-19 on brain structure would overlap with PTSD. The topic is very interesting for readers, aims are clear and results are presented in a quantitative and structured way. All sections of manuscripts cover with the scope of the journal. However, some recommendations are suggested in order to enhace improvements in the current form of this manuscript.

Thank you for your positive attitude. Please find the changes with respect to the last version in red characters; deletions will not appear.

Methods

1)      Did you register the review as a protocol -if not this needs to be added to your limitations.

We did not, but it wasn’t our fault. We tried registering with Prospero, but after having reached the end of the submission, the York University site remained idle for hours. We tried it twice, when we tried to save it the site tilted and went idle. Sincerely, we don’t feel like to try again. We don’t feel either that our registration could have increased the quality of our paper. Aty any rate, we added the lack of registration to the limitations of our paper, although we don’t believe it’s a real limitation, it’s just bureaucracy.

2)      For the search strategy can you add how many people performed the search and if the processess were undertaken blind.

Three independent researchers performed the searches. Since the search was agreed upon in encounters among participant researchers, it could not be blind, but each moved independent from one another. It should be clear after carefully reading our Methods section. There we wrote “Three authors independently conducted the agreed search and compared their results” and then described that we reached absolute consensus through Delphi rounds. Hence, we could not add anything else to our Methods that was not written before.

3)      Please consider the risk of bias, effect measures, reporting bias assessment and certainly assesment items with are in the PRISMA 2020 checklist.

We did not assess risk-of-bias (RoB) in our review, as the methods of the eligible studies were so heterogeneous to render one RoB assessment tool to apply for all. We did not deal with this issue specifically in the manuscript but added this to the Limitations section.

Results

It would be betther if authors xould include the risk of bias assessment e.g AXIS tool for cross sectional studies and certainly assessment. It would improve the quality of results.

At your suggestion, we used the tool for cross-sectional studies. Actually, it’s not a RoB tool, but may assess quality and selection bias. We provided a detailed description in the Supplement and corrected our PRISMA Statement Checklist accordingly. We added to Methods and Limitations.

Despite this recommendation, from my point of view the core of the article is important and relevant and I suggest the paper has a huge potential for publication.

Thank you for the overall impression and for the precious suggestions that helped us improving considerably our manuscript.

Reviewer 2 Report

The paper discusses Are the Post-COVID-19 Posttraumatic Stress Disorder (PTSD) Symptoms Justified by the Effects of COVID-19 on Brain Structure? The "brain fog" that COVID-19 patients frequently experience and some cognitive impairment shown in many patients in the post-COVID-19 era indicate that COVID-19 impacts brain function. Even after fully recovering from the acute somatic illness, about one-third of individuals still exhibit posttraumatic stress disorder (PTSD) symptoms. We predicted that the long-lasting modifications to brain structure brought on by COVID-19 would coincide with those connected to PTSD. The paper has a good flow and provides enough info for the reader to get by.

Suggestion / Revision:

1.      The abstract must be more precise about the problem statement and the author's contribution.

2.      The authors must separate the introduction from the literature survey.

3.      The research motivation must be mentioned at the end of the introduction section.

4.      Authors must confirm that all acronyms are defined before being used for the first time. 

5.      The result section can be improved using some more description.

6.      Authors are suggested to proofread the manuscript after addressing all comments to avoid any typos, grammatical and lingual mistakes, and errors

Authors are suggested to proofread the manuscript after addressing all comments to avoid any typos, grammatical and lingual mistakes, and errors

Author Response

The paper discusses Are the Post-COVID-19 Posttraumatic Stress Disorder (PTSD) Symptoms Justified by the Effects of COVID-19 on Brain Structure? The "brain fog" that COVID-19 patients frequently experience and some cognitive impairment shown in many patients in the post-COVID-19 era indicate that COVID-19 impacts brain function. Even after fully recovering from the acute somatic illness, about one-third of individuals still exhibit posttraumatic stress disorder (PTSD) symptoms. We predicted that the long-lasting modifications to brain structure brought on by COVID-19 would coincide with those connected to PTSD. The paper has a good flow and provides enough info for the reader to get by.

Thank you for the appreciation. Please find the changes from the last version in red letters in the manuscript; deletions won’t appear.

Suggestion / Revision:

  1. The abstract must be more precise about the problem statement and the author's contribution.

Thank you for the suggestion. We added the possible contribution to future study designs.

  1. The authors must separate the introduction from the literature survey.

Thank you for the suggestion. We created a new subsection to comprise the literature survey.

  1. The research motivation must be mentioned at the end of the introduction section.

We mentioned the research motivation at the end of the introduction section.

  1. Authors must confirm that all acronyms are defined before being used for the first time.

Thank you for reminding us, we found no errors in this.

  1. The result section can be improved using some more description.

Thank you for your suggestion; we expanded the results section by adding more details.

  1. Authors are suggested to proofread the manuscript after addressing all comments to avoid any typos, grammatical and lingual mistakes, and errors

Thank you for this suggestion, we reviewed carefully the entire manuscript and the online Supplement for possible typos, misprints, and language errors, but we are mother tongue English.

Authors are suggested to proofread the manuscript after addressing all comments to avoid any typos, grammatical and lingual mistakes, and errors

This is the same question as above, we repeat that we reviewed carefully the entire manuscript and the online Supplement for possible typos, misprints, and language errors, but we are mother tongue English and the few misprints or syntax errors we found, we corrected them. Thank you for all suggestions that helped us improving our manuscript and hope you will endorse the new version.

Reviewer 3 Report

The manuscript is a systematic review that aims to clarify if persistent changes induced by COVID-19 on brain structure would overlap with those associated with PTSD. It follows PRISMA guidelines, and the systematic review procedure is well-designed and implemented. However, I have some concerns.

My may concern and maybe an advantage (when making it clearer and more explicit) is that it is not a standard review. I was not aware of it almost until the discussion. Normally, a review includes articles that have better or worst directly study the main aim of the review. In this case, for this review, the overlap between COVID-19 and PTSD brain structures is not directly studied in any of the included articles, except for one. Therefore, the article in this sense has a novelty not so common in systematic reviews, but it also implies limitations, since this issue has been never directly investigated, many confounding factors might be affecting the conclusions. Nonetheless, I think this might be a good first step for future research.

Introduction: please, better explain why is it relevant to know if the same structures might be implicated in PSTD and COVID.

Abstract: Please explain briefly why the hypothesis is partly supported. Which structure overlaps? What is missing?

Results: A whole metanalysis is probably not possible, but maybe it is possible to include various metanalysis grouping articles using, for example, similar techniques and related aims, that would provide nice information.

Discussion: Some of the structures that seem to be overlapped as the frontal cortex, ACC, and insula, are also involved in most emotional alterations (anxiety depression) and chronic pain. The alteration of these structures in Covid-19 patients might then be due to other psychological alterations rather than PTSD. To me the main conclusion of this review is that further research is needed, taking into account not only PTSD but other psychological factors in the covid-19 patients, since with the current data it is hard to arrive at any conclusion. Even the authors are aware of it since the conclusion is somehow inconclusive (partial support).

Please, provide future directions and how this review can help.

Author Response

The manuscript is a systematic review that aims to clarify if persistent changes induced by COVID-19 on brain structure would overlap with those associated with PTSD. It follows PRISMA guidelines, and the systematic review procedure is well-designed and implemented. However, I have some concerns.

Thank you for your positive attitude.

My may concern and maybe an advantage (when making it clearer and more explicit) is that it is not a standard review. I was not aware of it almost until the discussion. Normally, a review includes articles that have better or worst directly study the main aim of the review. In this case, for this review, the overlap between COVID-19 and PTSD brain structures is not directly studied in any of the included articles, except for one. Therefore, the article in this sense has a novelty not so common in systematic reviews, but it also implies limitations, since this issue has been never directly investigated, many confounding factors might be affecting the conclusions. Nonetheless, I think this might be a good first step for future research.

Thank you for this observation. It was indeed our aim to carry-out an atypical review using the standard reviews’ criteria.

Introduction: please, better explain why is it relevant to know if the same structures might be implicated in PSTD and COVID.

Thank you for this suggestion. We added why in Introduction.

Abstract: Please explain briefly why the hypothesis is partly supported. Which structure overlaps? What is missing?

We added in the Abstract which structures and functions overlapped, without adding what is missing, as everything else is missing.

Results: A whole metanalysis is probably not possible, but maybe it is possible to include various metanalysis grouping articles using, for example, similar techniques and related aims, that would provide nice information.

Thank you for the suggestion, but even studies using similar techniques and aims had differences (for example, use of the Montréal Neurological Institute vs. the Harvard Axis vs. Talairach-Tournoux) that did not allow us the simplest of the meta-analyses.

Discussion: Some of the structures that seem to be overlapped as the frontal cortex, ACC, and insula, are also involved in most emotional alterations (anxiety depression) and chronic pain. The alteration of these structures in Covid-19 patients might then be due to other psychological alterations rather than PTSD. To me the main conclusion of this review is that further research is needed, taking into account not only PTSD but other psychological factors in the covid-19 patients, since with the current data it is hard to arrive at any conclusion. Even the authors are aware of it since the conclusion is somehow inconclusive (partial support).

Thank you for this suggestion. The emotional alterations you mentioned, at least some of them, might have a traumatic origin (childhood abuse and neglect), so it would be impossible to disentangle them. We stressed this important concept in limitations.

Please, provide future directions and how this review can help.

We put that in the Conclusions section. We thank you for your useful suggestions that helped us improving the quality of our manuscript.

Round 2

Reviewer 3 Report

The authors have considered all my recommendations, and the manuscript is my opinion quite improved